# Measurement of the Imaginary Part of the Clausius-Mossotti Factor of Particle/Cell via Dual Frequency Electrorotation

**DOI:** 10.3390/mi11030329

**Published:** 2020-03-22

**Authors:** Yung-Yi Lin, Ying-Jie Lo, U Lei

**Affiliations:** 1Nanometrics Incorporated, Milpitas, CA 95035, USA; ylin@nanometrics.com; 2Institute of Applied Mechanics, National Taiwan University, Taipei 10617, Taiwan; d93543008@ntu.edu.tw

**Keywords:** imaginary part of the Clausius-Mossotti factor, electrorotation, travelling wave dielectrophoresis, dielectrophoresis, dual frequency

## Abstract

A simple and inexpensive method using planar electrodes was proposed for the measurement of the imaginary part of the Clausius-Mossotti factor, $K_{i}$, of particle/cell for electrorotation (ER) and travelling wave dielectrophoresis (twDEP). It is based on the balance between the dielectrophoretic and viscous torques on a particle undergoing ER subject to dual frequency operation in an ER chamber. A four-phase ac voltage signal with a given frequency is applied for generating ER for measurement, and another two-phase signal is applied at a selected frequency for generating a negative dielectrophoretic force for confining the particle motion, instead of using laser tweezer or three-dimensional electrodes in the literature. Both frequencies can be applied to the same electrodes in a four-electrode ER system and to alternative different electrodes in an eight-electrode ER system, and both systems are capable for providing accurate measurement. The measurements were validated by comparing with the theoretical result using sephadex particles in KCl solution, and with the existing experimental results for various human cancer cells in medium with conductivity from 0.01–1.2 S/m, using ER with optical tweezer and dual frequency twDEP. Contrast between the ER and the twDEP methods (the current two available methods) was discussed and commented. The present method could provide measurement for wider frequency range and more accurate result near $K_{i}$ = 0, in comparison with the results using the twDEP method. However, the twDEP method could perform much more rapid measurement. Detailed forces and torque were calculated inside the ER chamber for understanding the physics and assessing the characteristics of the dual frequency ER method. This study is of academic interest as the torque in ER and the force in twDEP can be calculated only when $K_{i}$ is known. It also finds biomedical applications as the $K_{i}$-spectra can be served as physical phenotypes for different cells, and can be applied for deriving dielectric properties of cells.

## 1. Introduction

AC electrokinetics, including conventional dielectrophoresis (DEP), electrorotation (ER), and travelling wave dielectrophoresis (twDEP), are effective tools for the manipulation and characterization of particles and cells [1,2,3,4], as noncontact force and/or torque can be exerted to the particle through the application of an appropriate ac electric field. A particle moves in a non-uniform electric field when it is subject to a conventional dielectrophoretic force, rotates in a constant rotating electric field when it is subject to a dielectrophoretic torque, and experiences both dielectrophoretic force and torque in a travelling wave electric field generated by an array of electrodes with phase shift between neighboring electrodes, and the associated phenomena are called the DEP, the ER and the twDEP, respectively, in the literature. Details of DEP can be found from Ref. [1,2,3,4], an introduction to ER can be found from Ch. 4 of Ref. [2], and the fundamentals and applications of twDEP can be found from Ch. 8 of Ref. [3] and Chapters 10 and 11 of Ref. [4]. 

The force and torque associated with twDEP and the torque associated with ER depend on a combined parameter, $K_{i}$, the imaginary part of the Clausius-Mossotti factor. For a homogeneous spherical particle with uniform conductivity $\sigma_{p}$ and permittivity $\varepsilon_{p}$ in a medium with uniform conductivity $\sigma_{m}$ and permittivity $\varepsilon_{m}$ [5],
(1)$$K_{i}=\frac{\omega \left(\varepsilon_{p}-\varepsilon_{m}\right)\left(\sigma_{p}+2\sigma_{m}\right)-\omega \left(\sigma_{p}-\sigma_{m}\right)\left(\varepsilon_{p}+2\varepsilon_{m}\right)}{\omega^{2}{\left(\varepsilon_{p}+2\varepsilon_{m}\right)}^{2}+{\left(\sigma_{p}+2\sigma_{m}\right)}^{2}}$$
where ω is the angular frequency of the applied ac electric field. For many practical situations, such as the applications involving cells, the particles are inhomogeneous; $\sigma_{p}$ and $\varepsilon_{p}$ in Equation (1) could be replaced by their equivalent values using the layered model [2,3,4,6]. However, the permittivity and conductivity of each layer (say, the cell membrane and the cytoplasm) are still unknown in general, and thus Equation (1) cannot be applied. Therefore, it would be helpful if $K_{i}$ could be measured directly. The knowledge of $K_{i}$ is fundamental from the academic point of view, as we can calculate the torque and the force exerted on the particle for the ER and twDEP applications only when we know $K_{i}$, which is of particular interest for the theoretical researchers. The role of $K_{i}$ in ac-electrokinetics (ER and twDEP) is similar to the role of Young Modulus in Elasticity and viscosity in Newtonian fluid mechanics. 

Variation of $K_{i}$ with frequency, called the $K_{i}$-spectrum, for TentaGel particles and Jurkat cells were determined via ER in a three-dimensional, asymmetric octopole micro cage [7,8]; the $K_{i}$-spectra for yeast cells, Jurkat cells and red blood cells were also determined via ER in a symmetric three-dimensional octopole micro cage with the aid of an optical tweezer [9] for particle confinement in case of positive DEP. The $K_{i}$-spectra of human colorectal cancer cell, Colo 205, in medium with conductivity from 0.011 to 1.1 S/m were measured using ER generated by planar electrodes with an optical tweezer [10]. Recently, a twDEP method was proposed [5] for measuring $K_{i}$ based on the force balance of a constant moving cell in a travelling wave electric field inside a designed micro channel, and the $K_{i}$-spectra for Colo205, as well as two human lung cancer cells, CL1-0 and CL1-5, were measured, in physiological buffer (with medium conductivity at 1.6 S/m) where the cells behave negative DEP. In case of positive DEP, such as that in a DEP buffer with conductivity from 0.1 to 0.01 S/m, dual frequency operation was employed for the twDEP measurement [11], for avoiding the adhesion of cells on electrodes. 

Based on the above, there are two methods, the ER method and the twDEP method, available in the literature for the measurement of $K_{i}$; and each of them has its own advantage and disadvantage. The twDEP method has the advantage over the ER method that it is simpler and more effective; measurement is performed in a flow-through manner and analytical solution is available for the electric field, provided the particle is translating at an appropriate height from the electrode array under suitable design [12]. On the other hand, cell-by-cell measurements were performed, and numerical solution of the electric field is required for the ER method. However, the measurement using twDEP could be inaccurate in comparison with that using ER when $K_{i}\approx 0$, as shown from the results in Ref. [11]. In particular, the measured values of $K_{i}$ are essentially zero for a range of frequency using twDEP, instead of $K_{i}=0$ at a single frequency within the associated frequency range for most of the ER measurements. Also, the lower limit of the frequency range of the twDEP measurement is higher than that of the ER measurement, result in a shorter frequency range of data obtained using the twDEP method. Thus, the ER measurement is a better choice if a detailed knowledge of the $K_{i}$-spectrum is required, for example, in case one wishes to derive the cell dielectric properties using the imaginary part of the Clausius-Mossotti factor. Regarding the current ER measurements, it would be much helpful if it can be performed without using optical tweezer or electrode cages for particle confinement. Therefore, the primary goal of the present study is to develop a modified ER method, with dual frequency operation and planar electrodes (as those in conventional ER), as will be further discussed. Measurements will be performed for studying $K_{i}$-spectra of different cells in medium with different conductivities, ranging from 0.01–1.2 S/m. The idea of using dual frequency operation in ac-electrokinetics is not new, as the operation was applied successfully in twDEP [11,13] and in ER [14] systems, and the performance of the particle manipulation can be enhanced [15].

The so-called electrorotation (ER, or ROT) spectra, which express the variations of the rotation speed Ω of the particle with the applied electric frequency (f = ω/(2π)), were commonly measured in the literature [16,17,18,19,20,21,22,23,24] in comparison with the $K_{i}$-spectra. The $K_{i}$-spectra and the ER-spectra are similar, but they are not the same. They are related by
(2)$$\Omega =-0.5K_{i}\varepsilon_{m}E^{2}/\eta$$Here E is the magnitude of the electric field where the particle locates, and η is the viscosity of the medium. The ER spectrum and the $K_{i}$-spectrum are linearly proportional to each other; both can be served as physical phenotypes of cells, and applied for deriving the dielectric properties of cells. However, there are some differences between those two spectra as follows. (i) The ER spectrum depends on E, and is thus device dependent, whereas $K_{i}$ is device independent. One can compare quantitatively the $K_{i}$-spectra, but not the ER spectra, measured from different devices. (ii) Furthermore, the values of Ω at different frequencies in the ER–spectrum refer to situations of different particle heights [9] from the substrate in general for a suspended particle, as the DEP force acting on the particle is different for different frequency due to the different value of the real part of the Clausius-Mossotti factor. As a result, the electric field (which depends on particle location), and thus the dielectrophoretic torque experienced by the particle are different for different data points in the ER–spectrum; or, the values of Ω at different frequencies in an ER–spectrum are generated by different torques. If one wishes to convert the ER-spectrum to the $K_{i}$-spectrum (or vice versa) using Equation (2), E should be of different values for different Ω’s because the latter is position dependent. (iii) However, the electric field magnitudes (or the dielectrophoretic torques) are generally assumed to be the same for all the data points in the ER–spectrum when those data were applied for deriving the dielectric properties of particle in the literature. With the derived dielectric properties, the $K_{i}$-spectrum can be calculated using Equation (1). The $K_{i}$-spectra thus constructed could be different (less accurate) from those through direct measurements in the literature [7,8,9,10] and also in the present study, as the effect of electric field variation was included in the direct measurements.

In order to perform steady and robust ER measurement, the test particle needs to stay in a mechanically stable position with essentially constant dielectrophoretic torque during the process of measurement in the ER device. The conventional DEP force acting on the test particle is the primary force involved in the mechanical balance, and thus the knowledge of the real part of the of the Clausius-Mossotti factor [5],
(3)$$K_{r}=\frac{\left(\varepsilon_{p}-\varepsilon_{m}\right)\left(\varepsilon_{p}+2\varepsilon_{m}\right)\omega^{2}+\left(\sigma_{p}-\sigma_{m}\right)\left(\sigma_{p}+2\sigma_{m}\right)}{\omega^{2}{\left(\varepsilon_{p}+2\varepsilon_{m}\right)}^{2}+{\left(\sigma_{p}+2\sigma_{m}\right)}^{2}}$$
is also required in the determination of $K_{i}$ using ER. The time average conventional dielectrophoretic (DEP) force on the particle [2],
(4)$$F_{c}=2\pi \varepsilon_{m}R^{3}K_{r}\nabla E_{rms}^{2}$$
which is related directly to $K_{r}$. Here R is the radius of the particle (assumed spherical), and $E_{rms}^{2}$ is the mean square of the electric field magnitude where the particle locates. When $K_{r}$ > 0 (called positive DEP), $F_{c}$ is in the direction of $\nabla E_{rms}^{2}$; the particle tends to move toward the edges of the electrodes and adheres there in the ER chamber, and the ER stops. When $K_{r}$ < 0 (called negative DEP), $F_{c}$ is opposite to the direction of $\nabla E_{rms}^{2}$; the particle tends to move away from the edges of the electrodes to a location of relative minimum value of $E_{rms}^{2}$ in the ER chamber, i.e., the central region of the chamber. The particle rotates steadily there, and the ER measurement is performed. The particle is mechanically stable in a position on the vertical axis of the ER chamber for negative DEP (provided it is not too close to the top wall of the chamber), and is unstable for positive DEP. In case of positive DEP, an additional confinement force is required for overcoming the DEP force, such as the optical force ([9,10]), or others, for holding the particle at an essentially fixed position near the chamber axis. Here we propose a negative DEP force corresponding to a signal with selected voltage and frequency, which is applied simultaneously with the signal for generating the electrorotation. There are some ER-spectra measurements for titanium [25] and ZnO [26] particles in the literature without the use of additional means for particle confinement (optical tweezer, electrode cage, and dual frequency operation as mentioned above). A possible explanation is that the radial outward movement of the particle associated with positive DEP (if exists) in those studies was inhibited by the friction between the particle and the substrate; the friction plays the role as a confining force. The test particles are too heavy, with densities (about 4.51 g/cm^3^ for titanium in [25] and 5.61 g/cm^3^ for ZnO [26]) much greater than that of the surrounding medium (around 1 g/cm^3^), such that they settle on the substrate during the measurement; also the particle inertia is large. On the other hand, the cells are just slightly denser than the medium by about 5%; they are usually suspended during the measurement in practice, and thus can escape easily from the test region of the ER chamber under positive DEP, as perturbations always exist in the system. As for the $K_{i}$ measurement of cells here, it is preferred to perform the test in suspended state in order to avoid any additional adhesion surface forces between the cell and the substrate [27]. Those adhesion forces cannot be expressed easily in analytical forms, and will complicate the balance between the dielectrophoretic torque and viscous torque, the theoretical basis for $K_{i}$ measurement. Thus, the test particle, particular for cell, needed to be confined via some means under positive DEP in the $K_{i}$ measurement using ER, which were employed and discussed in the literature [7,8,9,10].

The experimental studies on $K_{r}$ are limited in the literature. Values of $K_{r}$ for colloidal particles [28] and engineered protein patterned colloidal particles [29] in low conductivity medium of order 10^−4^ S/m were measured, by balancing the DEP force with the viscous drag in a rectangular micro channel using the electric field generated by actuating two parallel electrodes on the channel substrate. Several human cancer cells [5] were measured in medium with conductivity from 0.01 to 1.1 S/m, by also balancing the DEP force with the viscous drag on the particle, but in a designed radial electric field in a micro channel such that analytical solution for the electric field exists. There is also a method, called the isomotive dielectrophoresis [30,31,32], could be applied for the measurements of $K_{r}$. The measurements based on force balance according to the Newton’s second law above are valid for particles (cells) greater than about 5 μm, but statistical mechanics issue, such as the Smoluchowski equation, is required for taking into account the Brownian effects for measuring $K_{r}$ of sub-micron particles [33]. The results for $K_{r}$ of Ref. [5] will be employed here for designing the confining force in the present ER measurement.

## 2. Materials and Methods

Both experiments and theoretical calculations (for supporting the experimental design and understanding the physical reasoning) were performed in this study. 

### 2.1. Device

The device, as shown in Figure 1a, was modified from the device of Ref. [10]. It is an electrorotation chamber (ER chamber) with gold electrodes deposited on its glass substrate (with a chrome layer in between for improving the adhesion), and its top and side walls were molded with polydimethylsioxane (PDMS), using standard MEMS techniques, including photolithography, wet etching, and molding using PDMS. Details of the methods and techniques are available from chapter 9 of Ref. [3] and Ref. [34]. The modification made here from Ref. [10] is that dual frequency signals were applied to the electrodes, instead of using an optical tweezer in [10], for confining the ER of cell locally above the center of the electrodes. There are two options for the application of dual frequency in this study, the four-electrode version and the eight-electrode version, as shown in Figure 1b,c, respectively, with length scales (electrode width and tip-to-tip spacing) and phases of the applied electric signals indicated.

Let $V_{4}cos(\omega_{4}t+\varphi_{4e})$ be the voltage signal (called the 4-phase signal here) applied to the electrodes in Figure 1a for generating electrorotation, with $V_{4}$ the amplitude, $\omega_{4}$ the angular frequency, $\varphi_{4e}$ the phase, and t the time. There is a $90^{o}$ phase shift between neighboring electrodes in the present study, i.e., $\varphi_{4e}=0^{o},\;90^{o},\;180^{o}\;or\;270^{o},$ respectively, as indicated in those four electrodes in Figure 1a. $K_{i}$ is measured for a given value of $\omega_{4}$ if the test particle performs steady electrorotation, which would occur if $\omega_{4}$ is a frequency that the particle behaves negative DEP. 

In case of positive DEP, another ac voltage signal (called the 2-phase signal here), $V_{2}cos(\omega_{2}t+\varphi_{2e})$, with voltage $V_{2}$, frequency $\omega_{2}$ and phase $\varphi_{2e}$ ($=0^{o}$ or $180^{o}$, with $180^{o}$ phase shift between neighboring electrodes), is also applied to the electrodes. $V_{2}$ and $\omega_{2}$ are chosen such that a negative DEP force is generated, which overcomes the positive DEP force associated with $\omega_{4}$. Both the 4-phase and the 2-phase signals can be applied simultaneously to the same electrodes as those in Figure 1b, or applied to different electrodes as those in Figure 1c, with the 4-phase signals indicated in red and the 2-phase signals in black in the figures. The area of each electrode is a rectangle with a semicircle at one of its end, and the electrodes are placed symmetrically around a circle on the substrate. The width of the 4-electrode system (100 μm) in Figure 1b is twice that of the 8-electrode system (50 μm) in Figure 1c, and the tip-to-tip distances of the opposite electrodes, s, for both systems are 100 μm. The ER chamber is fabricated in the central part of a long rectangular channel (with width 1000 μm), with obstacles built on both sides, as shown in Figure 1d. The obstacles were placed for damping out possible flow oscillation during the experiment [10]. Figure 1e shows an enlarged view of the test region for the eight-electrode system, together with a human lung cancer cell, CL1-5, which is one of the test cells in this study. The present method was also validated using Equation (1) via the measurement of sephadex particle (G-25 super fine, GE Healthcare Life Science) in KCl solution. The sephadex particle is well established gel filtration resin for desalting and buffer exchange in industrial applications; it possesses known dielectric properties. The purchased particles are poly-dispersed (from 15–88 μm), and some selected particles are shown in Figure 1f. Particles with diameters between 20–30 μm were chosen for experiment, and such a size range is consistent with the sizes of the cells in this study. Some details on sephadex particles are available in Ref. [35]. Figure 1g shows four snapshots from a video recorded for an ER experiment, which will be further discussed.

### 2.2. Theory

Let $Re\left\{\left[\Phi_{4r}(x,y,z)+j\Phi_{4i}(x,y,z)\right]exp\left(j\omega_{4}t\right)\right\}$ and $Re\left\{\Phi_{2}(x,y,z)exp\left(j\omega_{2}t\right)\right\}$ be the electric potentials in the fluid medium inside the ER chamber corresponding to the applied 4-phase and 2-phase signals, respectively, with (x,y,z) the coordinates as shown in Figure 1, Re{…} the real part of {…}, and $j=\sqrt{-1}$. The electric potential functions, $\Phi_{4r}(x,y,z)$, $\Phi_{4i}(x,y,z)$ and $\Phi_{2}(x,y,z)$, are all real, and could be obtained numerically by solving the Laplace equations according to electrostatics, subject to the associated specified potentials on the electrodes, insulated conditions on the PDMS walls and glass substrate, and zero gradient of potentials at the other (“outlet”) boundaries of the calculation domain [6,36]. It follows that the corresponding electric fields,
(5)$$E_{4}=-\nabla Re\left\{\left[\Phi_{4r}(x,y,z)+j\Phi_{4i}(x,y,z)\right]exp\left(j\omega_{4}t\right)\right\}$$
and
(6)$$E_{2}=-\nabla Re\left\{\Phi_{2}(x,y,z)exp\left(j\omega_{2}t\right)\right\}=-\nabla \left\{\Phi_{2}(x,y,z)cos\left(\omega_{2}t\right)\right\}$$

The quasi-static electric fields were usually expressed as [6,36,37]
(7)$$E_{4}\left(x,y,z,t\right)=E_{4x}cos\left(\omega_{4}t+\varphi_{x}\right)\hat{x}+E_{4y}cos\left(\omega_{4}t+\varphi_{y}\right)\hat{y}+E_{4z}cos\left(\omega_{4}t+\varphi_{z}\right)\hat{z}$$
and
(8)$$E_{2}\left(x,y,z,t\right)=\left(E_{2x}\hat{x}+E_{2y}\hat{y}+E_{2z}\hat{z}\right)cos\omega_{2}t$$
where $\left(\hat{x},\hat{y},\hat{z}\right)$ are the unit vectors in the (x, y, z) coordinates system, with
(9)$$\begin{array}{l} E_{4x}=\sqrt{{\left(\frac{\partial \Phi_{4r}}{\partial x}\right)}^{2}+{\left(\frac{\partial \Phi_{4i}}{\partial x}\right)}^{2}},\\ E_{4y}=\sqrt{{\left(\frac{\partial \Phi_{4r}}{\partial y}\right)}^{2}+{\left(\frac{\partial \Phi_{4i}}{\partial y}\right)}^{2}},\\ E_{4z}=\sqrt{{\left(\frac{\partial \Phi_{4r}}{\partial z}\right)}^{2}+{\left(\frac{\partial \Phi_{4i}}{\partial z}\right)}^{2}}, \end{array}$$
(10)$$\begin{array}{l} \varphi_{x}=sin^{-1}\left(\frac{-\partial \Phi_{4i}/\partial x}{E_{4x}}\right),\\ \varphi_{y}=sin^{-1}\left(\frac{-\partial \Phi_{4i}/\partial y}{E_{4y}}\right),\\ \varphi_{z}=sin^{-1}\left(\frac{-\partial \Phi_{4i}/\partial z}{E_{4z}}\right), \end{array}$$
and
(11)$$\begin{array}{l} E_{2x}=-\frac{\partial \Phi_{2}}{\partial x},\\ E_{2y}=-\frac{\partial \Phi_{2}}{\partial y},\\ E_{2z}=-\frac{\partial \Phi_{2}}{\partial z}, \end{array}$$
obtained by comparing Equations (5) and (6) with (7) and (8). The time average total force acting on a spherical particle subject to dual frequency operation can then be evaluated as [11]
(12)$$F_{t}=F_{c4}+F_{tw4}+F_{c2}$$
where
(13)$$F_{c4}=2\pi \varepsilon_{m}R^{3}K_{r4}\nabla E_{rms4}^{2}$$
(14)$$F_{tw4}=2\pi \varepsilon_{m}R^{3}K_{i4}\left(E_{4x}^{2}\nabla \varphi_{x}+E_{4y}^{2}\nabla \varphi_{y}+E_{4z}^{2}\nabla \varphi_{z}\right)$$
and
(15)$$F_{c2}=2\pi \varepsilon_{m}R^{3}K_{r2}\nabla E_{rms2}^{2}$$
are the conventional DEP force associated with the 4-phase signal, the twDEP force associated with the 4-phase signal, and the conventional DEP force associated with the 2-phase signal, respectively, with $E_{rms4}^{2}=0.5\left(E_{4x}^{2}+E_{4y}^{2}+E_{4z}^{2}\right)$, $E_{rms2}^{2}=0.5\left(E_{2x}^{2}+E_{2y}^{2}+E_{2z}^{2}\right)$, $K_{r4}$ the real part of the Clausius-Mossotti factor at frequency $\omega_{4}$, $K_{r2}$ the real part of the Clausius-Mossotti factor at frequency $\omega_{2}$, and $K_{i4}$ the imaginary part of the Clausius-Mossotti factor at frequency $\omega_{4}$. In the calculation here, the term $E_{4x}^{2}\nabla \varphi_{x}+E_{4y}^{2}\nabla \varphi_{y}+E_{4z}^{2}\nabla \varphi_{z}$ in Equation (14) was expressed in terms of the sums of products of differentials involving $\Phi_{4i}$ and $\Phi_{4r}$ using Equations (9) and (10) according to Ref. [36], so that the determination of the signs for the phase components in Equation (10) can be avoided. Figure 1h,i show the force balances of the particle when it is situated stably on the z-axis of the ER chamber for measurement. $F_{c4h}$ and $F_{c2h}$ ($F_{c4z}$ and $F_{c2z}$) are the horizontal (vertical) force components of $F_{c4}$ and $F_{c2}$, respectively, $F_{w}$ is the particle weight, and $F_{b}$ is the buoyancy of the particle, with the gravity **g** along the negative z-direction. The directions of $F_{c4h}$ and $F_{c4z}$ depend on the sign of $K_{r4}$, which depends on the applied frequency $\omega_{4}$. On the other hand, $F_{c2h}$ and $F_{c2z}$ are always pointing away from the electrodes, as $\omega_{2}$ is chosen to be a frequency for negative DEP. The density of the cell is slightly (about 5%) greater than that of the surrounding medium, and an upward net DEP force ($F_{c4z}$ + $F_{c2z}$ > 0) is required for suspending the cell for electrorotation. 

The dielectrophoretic torque for the dual frequency operation can also be derived using the same idea as that for deriving the force [11] in Equation (12). The effective dipole moments on the particle induced by the four-phase and two-phase electric fields in Equations (7) and (8) are [2,37]
(16)$$\begin{array}{ll} m_{4}= & 4\pi \varepsilon_{m}R^{3}\{E_{4x}\left[K_{r4}cos\left(\omega_{4}t+\varphi_{x}\right)-K_{i4}sin\left(\omega_{4}t+\varphi_{x}\right)\right]\hat{x}\\ & +E_{4y}\left[K_{r4}cos\left(\omega_{4}t+\varphi_{y}\right)-K_{i4}sin\left(\omega_{4}t+\varphi_{y}\right)\right]\hat{y}\\ & +E_{4z}\left[K_{r4}cos\left(\omega_{4}t+\varphi_{z}\right)-K_{i4}sin\left(\omega_{4}t+\varphi_{z}\right)\right]\hat{z}\} \end{array}$$
and
(17)$$\begin{array}{ll} m_{2} & =4\pi \varepsilon_{m}R^{3}\{E_{2x}\left[K_{r2}cos\omega_{2}t-K_{i2}sin\omega_{2}t\right]\hat{x}\\ & \;+E_{2y}\left[K_{r2}cos\omega_{2}t-K_{i2}sin\omega_{2}t\right]\hat{y}\\ & \;+E_{2z}\left[K_{r2}cos\omega_{2}t-K_{i2}sin\omega_{2}t\right]\hat{z}\}\\ & =4\pi \varepsilon_{m}R^{3}\left[K_{r2}cos\omega_{2}t-K_{i2}sin\omega_{2}t\right]\{E_{2x}\hat{x}+E_{2y}\hat{y}+E_{2z}\hat{z}\}, \end{array}$$
respectively. The electric fields and the effective dipole moments for the four-phase and the two-phase signals can be superimposed, as the potentials are governed by linear equation (Laplace equation). It follows that the dielectrophoretic torque [2]
(18)$$\begin{array}{ll} T & =m\times E=\left(m_{2}+m_{4}\right)\times \left(E_{2}+E_{4}\right)\\ & =\left(m_{2}\times E_{2}\right)+\left(m_{2}\times E_{4}\right)+\left(m_{4}\times E_{2}\right)+\left(m_{4}\times E_{4}\right). \end{array}$$ With Equations (7), (8), (16) and (17),
(19)$$m_{2}\times E_{2}=0$$
(20)$$\begin{array}{ll} m_{2}\times E_{4}= & 2\pi \varepsilon_{m}R^{3}\{\left(\hat{z}E_{2x}E_{4y}-\hat{x}E_{2z}E_{4y}\right)\left[K_{r2}\left(cos\left(\omega_{m}t+\varphi_{y}\right)+cos\left(\omega_{p}t+\varphi_{y}\right)\right)-K_{i2}\left(sin\left(\omega_{p}t+\varphi_{y}\right)-sin\left(\omega_{m}t+\varphi_{y}\right)\right)\right]\\ & +\left(\hat{x}E_{2y}E_{4z}-\hat{y}E_{2x}E_{4z}\right)\left[K_{r2}\left(cos\left(\omega_{m}t+\varphi_{z}\right)+cos\left(\omega_{p}t+\varphi_{z}\right)\right)-K_{i2}\left(sin\left(\omega_{p}t+\varphi_{z}\right)-sin\left(\omega_{m}t+\varphi_{z}\right)\right)\right]\\ & +\left(\hat{y}E_{2z}E_{4x}-\hat{z}E_{2y}E_{4x}\right)\left[K_{r2}\left(cos\left(\omega_{m}t+\varphi_{x}\right)+cos\left(\omega_{p}t+\varphi_{x}\right)\right)-K_{i2}\left(sin\left(\omega_{p}t+\varphi_{x}\right)-sin\left(\omega_{m}t+\varphi_{x}\right)\right)\right]\} \end{array}$$
(21)$$\begin{array}{cc} m_{4}\times E_{2}= & 2\pi \varepsilon_{m}R^{3}\{\left(\hat{x}E_{4y}E_{2z}-\hat{z}E_{4y}E_{2x}\right)\left[K_{r4}\left(cos\left(\omega_{m}t+\varphi_{y}\right)+cos\left(\omega_{p}t+\varphi_{y}\right)\right)-K_{i4}\left(sin\left(\omega_{p}t+\varphi_{y}\right)-sin\left(\omega_{m}t+\varphi_{y}\right)\right)\right]\\ & +\left(\hat{y}E_{4z}E_{2x}-\hat{x}E_{4z}E_{2y}\right)\left[K_{r4}\left(cos\left(\omega_{m}t+\varphi_{z}\right)+cos\left(\omega_{p}t+\varphi_{z}\right)\right)-K_{i4}\left(sin\left(\omega_{p}t+\varphi_{z}\right)-sin\left(\omega_{m}t+\varphi_{z}\right)\right)\right]\\ & +\left(\hat{z}E_{4x}E_{2y}-\hat{y}E_{4x}E_{2z}\right)\left[K_{r4}\left(cos\left(\omega_{m}t+\varphi_{x}\right)+cos\left(\omega_{p}t+\varphi_{x}\right)\right)-K_{i4}\left(sin\left(\omega_{p}t+\varphi_{x}\right)-sin\left(\omega_{m}t+\varphi_{x}\right)\right)\right]\} \end{array}$$
and
(22)$$m_{4}\times E_{4}=-4\pi \varepsilon_{m}R^{3}K_{i4}\{\hat{x}E_{4y}E_{4z}\sin\left(\varphi_{y}-\varphi_{z}\right)+\hat{y}E_{4z}E_{4x}\sin\left(\varphi_{z}-\varphi_{x}\right)+\hat{z}E_{4x}E_{4y}\sin\left(\varphi_{x}-\varphi_{y}\right)\}$$
where $\omega_{m}=\omega_{4}-\omega_{2}$ and $\omega_{p}=\omega_{4}+\omega_{2}$. The term $m_{2}\times E_{2}$ is identically zero as $m_{2}$ is parallel to $E_{2}$; the time averages of $m_{2}\times E_{4}$ and $m_{4}\times E_{2}$ are also zero, as all the terms in Equations (20) and (21) are periodic functions in time; the term $m_{4}\times E_{4}$ is the same as that in the literature for a single frequency ER [2,6]. Thus, there is no contribution to the time average dielectrophoretic torque from the electric field associated with the two-phase signal, and the dielectrophoretic torque acting on the particle for the dual frequency operation here is $m_{4}\times E_{4}$ in Equation (22), the same as that in the single frequency ER operation. The two-phase electric field does not contribute to the torque; it is introduced solely for providing a trapping force to the cell.

In particular, the DEP torque acting on the particle in the ER chamber for measuring $K_{i}$ (or more precisely, $K_{i4}$) is the z-component in Equation (22), which can be written as (see also [10])
(23)$$T_{DEP}=-4\pi \varepsilon_{m}R^{3}K_{i4}\left(\frac{\partial \Phi_{4i}}{\partial x}\frac{\partial \Phi_{4r}}{\partial y}-\frac{\partial \Phi_{4r}}{\partial x}\frac{\partial \Phi_{4i}}{\partial y}\right)$$
in terms of the electric potential functions [36].

Electric potentials, $\Phi_{4r}(x,y,z)$, $\Phi_{4i}(x,y,z)$ and $\Phi_{2}(x,y,z)$, were solved numerically, and the electric fields and the associated forces and torque in Equations (5)–(15) and (23) were then calculated for accessing the phenomena in the preset study, with the aid of the software COMSOL Multiphysics [38]. COMSOL Multiphysics is a general-purpose simulation software based on finite element method, capable for modeling multi-physics phenomena of many engineering and scientific problems. Here we employ it to solve the electric potentials and output the graphic results. With the potentials known, the associated electric fields were calculated using Equations (7)–(8) and (9)–(11), the forces using Equations (8) and (12)–(15), and the DEP torque using Equation (23). The associated quantities in Equations (7)–(15) and (23) were coded as self-defined functions in the software. Grid dependence tests were also performed for the calculation.

As shown in Figure 1j, the measurement of $K_{i4}$ here is based on the balance between the DEP torque and the viscous torque,
(24)$$-4\pi \varepsilon_{m}R^{3}K_{i4}\left(\frac{\partial \Phi_{4i}}{\partial x}\frac{\partial \Phi_{4r}}{\partial y}-\frac{\partial \Phi_{4r}}{\partial x}\frac{\partial \Phi_{4i}}{\partial y}\right)C_{DT}=8\pi \eta R^{3}\Omega C_{VT}$$
which was also employed previously in Ref. [10]. The left hand side of Equation (24) is $T_{DEP}$ in Equation (23) multiplied by a factor $C_{DT}$, accounting for the wall effect. Here the wall effects on both the DEP and viscous torques are included in Equation (24) for the experimental determination of $K_{i4}$, with $C_{DT}=1-3K_{r4}(\Delta_{1}^{3}+\Delta_{2}^{3})$ and $C_{VT}=1/(1-\Delta_{1}^{3}-\Delta_{2}^{3})$ the correction factors associated with the wall effects on DEP torque [39] and on viscous torque [40], respectively, and η is the viscosity of the fluid medium. In the above expressions, $\Delta_{1}=R/(2H)$, $\Delta_{2}=R/(2(H-h))$, H is the depth of the ER chamber, and h is the height of the test particle, as shown in Figure 1j. 

### 2.3. Experimental

Besides $\varepsilon_{m}$, η and R, three quantities are required in the calculation of $K_{i4}$ using Equation (24), and their determination are described as follows. (i) The electric potentials, $\Phi_{4i}$ and $\Phi_{4r}$, and their partial derivatives, in the ER chamber. They were calculated numerically as described in Section 2.2. (ii) The z (vertical) location of the particle from the substrate, h. It was estimated (within 2 μm uncertainty) through the differences between the scales of the focus screw of the microscope when it is focused at the particle and at the bottom wall of the chamber, respectively. The horizontal x and y locations of the particle, $x_{c}$ and $y_{c}$, can be measured easily from the image recorded through the microscope. With the particle location ($x_{c}$, $y_{c}$, h) known, the DEP torque on the left hand size of Equation (24), together with $C_{DT}$ and $C_{VT}$ can be evaluated. (iii) The average rotation speed Ω of the particle. It was evaluated based on the video recorded for the motion history of the electrorotation of the particle. It was calculated from the time spent for a given particle to perform a given numbers of revolution (usually five, other different numbers were checked). The numbers of revolution can be counted easily in the video for a cell as there always exist some characteristic marks on the image of the cell surface, as shown in Figure 1e. Figure 1g shows further that the orientations of the cell are different at different times. The video was recorded using a CCD camera incorporated with the microscope as that in Ref. [10]. For sephadex particles as those in Figure 1f, even the particle surface is quite smooth, the degree of darkness of the circular particle boundary in the image is not uniform (darker at top as in the figure, probably because that the incident light is not strictly parallel to the vertical particle axis). One can thus count the numbers of particle revolution using this characteristic, sometimes at a reduced playing rate of the video. In practice, the test particle (usually cell) may wobble slightly but periodically during the electrorotation, which might also be applied for counting the number of particle revolution. Three to five tests were performed for a case with a given set of the parameters in the present experiment. 

Measurement of $K_{i}$ (or $K_{i4}$) was validated first using sephadex particles (g-25 super fine, GE Healthcare Life Science) in KCl solution, with $\varepsilon_{P}/\varepsilon_{0}=66$ and $\sigma_{P}=0.0069\;S/m$ from Ref. [35] and $\varepsilon_{m}/\varepsilon_{0}=78$ and $\sigma_{m}=0.023\;S/m$ from measurement here, where $\varepsilon_{0}$ is the permittivity in vacuum. With properties known, $K_{i}$ can be calculated theoretically using Equation (1) for comparison. Validation was also performed by comparing the present measurements against previous measurements of various cells. Sephadex particles were employed before in ER measurements in [10,19] and twDEP measurements in [5,11].

Two human lung cancer cells, CL1-0 and CL1-5, and a colorectal cancer cell, Colo205, from cell lines, were employed for the experiments in this study. Those cells were employed previously in Ref. [5] for measuring $K_{r}$. The diameters of the cells were measured under a microscope. The mean diameters of CL1-0, CL1-5 cells and Colo205 cells are 16.8 μm, 15.9 μm and 14.5 μm, with standard deviations 3.2 μm, 2.5 μm and 1.5 μm, respectively. The cells were incubated in cultured medium, RPMI 1640, (Gibco BRL, Gaithersburg, MD) supplemented with 10% v/v feral bovine serum (FBS) and 1% v/v penicillin-streptomycin (Gibco BRL, Gaithersburg, MD). Two types of fluid medium were employed in the present study, the fresh cultured medium (RPMI solution) with conductivity 1.2 S/m, and the equal-osmotic mannitol solutions, made of 280 mM D-Mannit (Ferak, Germany) with their conductivities regulated by adding different amount of 3 M KCl solution. The adjusted conductivities of the mannitol solutions in the present study are 0.1 and 0.01 S/m. The cells were obtained from National Taiwan University hospital, and some detailed information about the cells and cell preparation were described in the appendix of Ref. [5].

The experimental parameters for the present study are listed in Table 1, including the medium conductivity $\sigma_{m}$; the applied voltage amplitude $V_{4}$ and frequency range $\omega_{4}$ for $K_{i}\left(or\;K_{i4}\right)$ measurement; the applied voltage amplitude $V_{2}$ and frequency $\omega_{2}$ for generating negative DEP force for particle confinement, together with the corresponding $K_{r2}$’s from Ref. [5]; and the measured settling heights of the particles/cells, h. The values with superscript * in the $K_{r2}$ column are values obtained through extrapolation. There is a range of h in the last column mainly because the values of $F_{c4}$ are different for different $K_{r4}$, which depends on $\omega_{4}$. A value h was measured for a given $\omega_{4}$. The values listed in Table 1 are for the eight-electrode system. The operation parameters for the four-electrode system, including the medium conductivities, the applied voltages and the applied frequencies are the same as those in Table 1, with the measured values of h different from those of the eight-electrode system.

## 3. Results and Discussion

### 3.1. Calculation of the Dielectrophoretic Forces and Torque in the ER Chamber

The dielectrophoretic forces, in particular the resulting trapping force $F_{th}$ (the horizontal component of $F_{t}$ in Equation (12)), and the vertical torque $T_{DEP}$ (Equation (23)), on the particle in the ER chamber will be examined via numerical calculation. Figure 2 shows the spatial distributions of dimensionless dielectrophoretic forces on a horizontal xy-plane at z = 40 μm in the ER chamber for a typical case under negative DEP (refer to the last row in Table 1); while Figure 3 shows the corresponding results for a typical case under positive DEP (refer to the eighth row in Table 1) on the plane at z = 32 μm. The origin of the plane at z = 40 μm (or 32 μm) is approximate the location where the cell settles stably and performs steady electrorotation for the corresponding negative (or positive) DEP case. ${\tilde{F}}_{c4h}$, ${\tilde{F}}_{tw4h}$, ${\tilde{F}}_{c2h}$, and ${\tilde{F}}_{th}$ in the figures are the dimensionless horizontal components of $F_{c4}$, $F_{tw4}$, $F_{c2}$, and $F_{t}$ in Equations (13), (14), (15) and (12), respectively, using force scale $F_{scale}(=2\pi \varepsilon_{m}R^{3}V_{4}^{2}/s^{3})$ for normalization, i.e., ${\tilde{F}}_{c4h}=F_{c4h}/F_{scale}$, with $F_{c4h}$ the horizontal component of $F_{c4}$, for example. Also indicated in the figures are the projections of the edges of the electrodes (shown as circular arcs in black) on the graphic plane (at z = 40 μm, or 32 μm), for horizontal position references. Note that the force vectors are pointing away from (or toward) these “circular arcs” (if the corresponding electrodes are actuated) for the negative (or positive) DEP case. 

Consider the left column of Figure 2 for the results of the four-electrode system of the negative DEP case. The trapping force at the central part of the ER chamber associated with ${\tilde{F}}_{c4h}$ in Figure 2a is rather weak (a large “blue” area at the center of the figure); it can be improved by superimposing a two-phase DEP force ${\tilde{F}}_{c2h}$ (Figure 2c) such that the resulting total force ${\tilde{F}}_{th}$ in Figure 2d shows a strong trapping force. This can be revealed from Figure 2d that there is a small blue (small force magnitude) circular area at the center, which is encircled by a thick red (large force magnitude) ring area, with force arrows pointing toward the center; the cell is trapped in the small blue area where ER is performed. The twDEP force, ${\tilde{F}}_{tw4h}$, is “circulating” (pointing along the azimuthal direction) around the center (Figure 2b), and is much less than ${\tilde{F}}_{c2h}$, as indicated by the maximum values shown at the upper right corners of the associated sub-figures. 

By comparing the results of the eight-electrode system on the right column with those of the four-electrode system on the left column in Figure 2, the total dimensionless horizontal trapping force, ${\tilde{F}}_{th}$, is stronger for the four-electrode system (maximum value, 0.283 versus 0.144). Although both the four-electrode and the eight-electrode systems have almost the same electrode areas in the ER chamber, both the 2-phase and the 4-phase signals apply to all electrodes for the four-electrode system, but either the 2-phase or the 4-phase signal applies to a given electrode in the eight electrode system, results in a larger trapping force for the four-electrode system. Anyway, ${\tilde{F}}_{th}$ of both systems here are strong enough for trapping the cells for the ER experiment.

For the positive DEP case in Figure 3, dual frequency operation is necessary as revealed from the directions of the arrows of ${\tilde{F}}_{c4h}$ (the DEP force for the 4-phase signal) in Figure 3a,e, which are pointing outward from the center. The total force ${\tilde{F}}_{th}$ in Figure 3d,h show that force traps can be formed at the center of the ER chamber by introducing a negative DEP force associated with the two-phase voltage signal (Figure 3c,g) in the operation. The horizontal trapping force ${\tilde{F}}_{th}$ for the four-electrode system is substantially stronger than that of the eight-electrode system; the trapping force is weak along the ±45° and ±135° directions for the eight-electrode system. As in Figure 2, the twDEP force ${\tilde{F}}_{tw4h}$ is “circulating” around the chamber center, and is substantially weaker than ${\tilde{F}}_{c4h}$.

Figure 4 shows the distributions of the horizontal dimensionless trapping force (i.e., ${\tilde{F}}_{th}$) on different planes at different heights (z = 10 μm, 25 μm, 40 μm and 55 μm) in the ER chamber (with height 80 μm), for the negative DEP case as that in Figure 2. The first, the second and the third rows correspond to the results for the four-electrode system with single frequency operation, the four-electrode system with dual frequency operation, and the eight-electrode system with dual frequency operation, respectively. The trapping force decreases rapidly as z increases for all these three cases. It is interest to see that for the present negative DEP case, the conventional DEP force for the single frequency operation indeed provides a trapping force at the center for small z’s, as shown in Figure 4a,b for z = 10 and 25 μm, respectively. However, the trapping force becomes weak at z = 40 μm, and the force even reverses its direction at z = 55 μm. This is due to the confinement of the electric field from the approximately insulated top surface. Thus, the particle cannot be trapped even for the negative DEP case if it is pushed to a sufficient large height in the ER chamber, which would occur when the applied voltage $V_{4}$ is too large. To avoid such a situation, dual frequency operation was applied here even for cases with negative DEP associated with $\omega_{4}$, and the particle can be trapped for both the four- and eight-electrode systems as shown in the results in Figure 4e–l. As the trapping force decreases with height, it would be better for choosing suitable parameters ($V_{2}$, $V_{4}$, $\omega_{2}$) for a given particle in a given medium in the ER chamber, such that the particle stays essentially near the middle horizontal plane (z = 40 μm here) of the chamber. This could be facilitated more easily with the aid of calculation. The parameters listed in Table 1 are suitable parameters thus chosen, and were demonstrated through experiments.

Figure 5a–h show the distributions of the vertical dimensionless DEP torque, ${\tilde{T}}_{z}\left(=T_{z}/T_{scale}\right)$, on different planes at different heights (z = 10 μm, 25 μm, 40 μm and 55 μm) in the ER chamber, for the negative DEP case as that in Figure 2, where $T_{scale}=4\pi \varepsilon_{m}R^{3}V_{4}^{2}/s^{2}$. The first and the second rows of Figure 5 show the results for the four-electrode and the eight-electrode systems, respectively. The torque decreases as z increases for both systems, as also shown in details in Figure 5i. However, the torques of the four-electrode system are 2.79–3.68 times (from the substrate at z = 0 to the top wall at z = h = 0.8 s) higher than those of the eight-electrode system. This is because of the same reason as that for explaining why the horizontal trapping force (${\tilde{F}}_{th}$) has a higher value for the four-electrode system in comparison with that for the eight-electrode system in Figure 2; both the four- and two-phase signals were applied to all the electrodes in the four-electrode system, whereas either one of the signals was applied to a given electrode in the eight-electrode system. Such a torque difference implies that the cell rotates more rapidly in the four-electrode system. The cell could sometimes be flung outward if it is rotating at a sufficiently high speed through the present experiment with suspended cells; thus, the eight-electrode system could provide a more stable environment for ER measurement in practice, as soon as the cell can be trapped stably.

The torque maximum on a given horizontal plane does not occur at the center when z is small (say, see the cases for z = 10 μm in Figure 5a,e); it does shift toward the center as z increases, and the variation of ${\tilde{T}}_{z}$ is small at the central part of the plane when z is sufficiently large (say, greater than 25 μm). The latter were also found in the calculations of Refs. [7,9]. The existence of an area with essentially constant DEP torque is important for ER measurement in practice, as some particles may wander slightly around the central axis of the chamber during electrorotation. In such a situation, the DEP torque in the torque balance equation (i.e., Equation (24)) still remains essentially constant throughout the measurement. However, the torque does vary substantially with z/s (or *h*/s, the dimensionless settling height of the particle in the experiment), as shown in Figure 5i,j.

Results of torque for the positive DEP case (say, the case in Figure 3) are qualitatively similar to those of the negative DEP case in Figure 5a–i, as can be seen by comparing Figure 5i with Figure 5j, which show the variations of dimensionless torques along the vertical central axis of the ER chamber. Although the torque values are quite different between Figure 5i,j, it is interested to find that the torque ratios between the 4-electrode system and the 8-electrode system are from 2.80 to 3.66 as z increases from 0 to 0.8 s in Figure 5j for the positive DEP case, which are almost the same as those in Figure 5i for the negative DEP case. 

In summary, numerical calculations were performed in this sub-section for examining the details of the trapping force and the DEP torque on the particle of the ER experiment, which are beneficial for understanding the mechanical behavior of the test particle inside the ER chamber, and for designing an effective and correct measurement of the imaginary part of the Clausius-Mossotti factor.

### 3.2. Experiment on the Imaginary Part of the Clausius-Mossotti Factor

The present method and device were validated first by comparing the experimental findings with the theoretical result according to Equation (1), using sephadex particle in KCl solution, with known dielectric properties $\varepsilon_{p}/\varepsilon_{0}=66$, $\sigma_{p}$ = 0.0069 S/m, $\varepsilon_{m}/\varepsilon_{0}=78$ and $\sigma_{m}$ = 0.023 S/m. For the present negative DEP case, the results agree with one other for those using the four-electrode system operated under single frequency, the four-electrode system operated under dual frequency, and the eight-electrode system operated under dual frequency, as shown in Figure 6. Both the four-electrode and the eight-electrode systems can be applied for the measurement of $K_{i}$. However, the low frequency limit of measurement for the four-electrode system under single frequency is 10^5^ Hz, which is higher than that under dual frequency operation, 10^3^ Hz. Larger frequency range of measurement can be obtained using dual frequency operation.

The present measurements were also validated by comparing with the existing experimental results using human colorectal cancer cell (Colo 205) and human lung cancer cells (CL1-0 and CL1-5), as shown in Figure 7 and Figure 8, respectively.

Consider first the Colo 205 cells in RPMI solution in Figure 7a. The cells exhibit negative DEP in the ER chamber. As in Figure 6 for the sephadex particles in KCl solution, the present results using both the four-electrode (operated under either single or dual frequency) and eight-electrode systems (operated under dual frequency) agree with one other in the measurement. The agreement between the results using single frequency and dual frequency for the four-electrode system implies that the dual frequency operation proposed here for the ER experiment is appropriate, as suggested by the theoretical study previously in the derivation of the time average DEP torque related to Equations (18)–(22). Figure 8a shows further validation using both the single and dual frequency ER in the four-electrode system with CL1-0 cells. Actually, manipulation using signal superposition in ac electrokinetics is rather common; there are successful studies using dual electric frequency signals in the literature, for examples, see References [11,13,14,15].

As the results using the eight-electrode system agree with those using the four-electrode system for the ER measurements according to Figure 6 and Figure 7a, and the measurement using the eight-electrode system is more stable in practice, the eight-electrode system will be employed mainly in this study.

Consider the ER result using the eight-electrode system under dual frequency operation here (the red points) and the ER result using four electrodes with optical tweezer (the green points) for particle confinement in Ref. [10], as shown in Figure 7. Both can be applied to study the cases with positive/negative DEP. The results for these two methods agree with each other essentially, indicating that both methods are adequate for the measurement of $K_{i}$. However, there are some discrepancies at some frequencies between both results.

Consider the data with discrepancies. Except for the limited results between 3 × 10^4^–10^6^ Hz in Figure 7c, the magnitudes of values of $K_{i}$ for the case using optical tweezer are less than those using dual frequency operation. This can be explained by the fact that the cells were suspended in the medium for experiments using dual frequency operation, while the cells were settled on the substrate for experiments using optical tweezer, during the ER measurements. Although the wall effects on the DEP and the viscous torque were included (see the factors $C_{VT}$ and $C_{DT}$ in Equation (24)) in the evaluation of $K_{i}$, the effect associated with the interfacial forces (such as van der Waals force and other adhesion force like ligand-receptor interaction, see Ref. [24]) was not accounted for. Such an interfacial effect is negligible in general when the cells are suspended at a distance of several microns from the substrate, but could be important when the cells are in contact with the substrate. It provides an additional resistance to the particle rotation, reducing its rotating speed, and thus the value of $K_{i}$. This is also one of the reasons why dual frequency operation was chosen for the ER measurement here, as the imposed negative DEP force associated with the two-phase frequency also provides a lifting force for suspending the test cell. The objective of the microscope incorporated with the optical tweezer employed in Ref. [10] has a NA (numerical aperture) value equals 0.55; the optical tweezer can only provide a lateral confining force to the cell and push it forward along the direction of the laser light, so that the cells in Ref. [11] settle on the substrate under an upright microscope. The optical tweezer could hold the test cell in suspended state if a higher NA (greater than unity) objective was employed. The discrepancy between the limited results in the frequency range 3 × 10^4^–10^6^ Hz in Figure 7c is, however, more likely due to different cell samples from cell cultures, although they are of the same cell type.

Consider the ER result using the eight-electrode system under dual frequency and the twDEP result under dual frequency for the Colo 205 cells in Figure 7 and the CL1-0 and CL1-5 cells in Figure 8. The agreements are good except the values of $K_{i}$ when they are near zero. For example, there are five data points near 10^7^ Hz in Figure 7b and seven data points near 10^6^ Hz in Figure 7c that $K_{i}=0$ in the twDEP results (the black data), but the curve for each ER result (the green or red data) in Figure 7b,c cuts across the line $K_{i}=0$ at a single point in the associated frequency range. It indicates that the ER method is more accurate than the twDEP method for measuring small values of $K_{i}$.

As the ER method and the twDEP method are two current methods available in the literature for measuring $K_{i}$, it is interesting to have a more detailed comparison between these methods. In the determination of $K_{i}$ (or $K_{i4}$) using ER according to Equation (24), it was determined by measuring the particle’s rotating speed Ω, with the electric potentials calculated. On the other hand, the twDEP electric field is generated by an array of parallel electrodes, with 90° phase shift between neighboring electrodes in general. The twDEP method for the determination of $K_{i}$ (or $K_{i4}$) is based on the force balance between $F_{tw4}$ and Stokes viscous drag along the direction of the electrode array [5,11],
(25)$$2\pi \varepsilon_{m}R^{3}K_{i4}\left(E_{4x}^{2}\frac{\partial \varphi_{x}}{\partial x}+E_{4z}^{2}\frac{\partial \varphi_{z}}{\partial x}\right)=6\pi \eta URC_{D}$$
where x is the direction along the electrode array with U the associated translating velocity component, z is the direction normal to the electrode array, and $C_{D}$ is the factor accounting for the wall effect on the viscous drag. $K_{i4}$ was determined by measuring the particle’s translating speed U, with the electric potentials, and thus the electric field and phase components, calculated numerically, or using analytical solution [12].

By comparing Equation (24) with (25), besides the accuracy of the measurement of Ω and U in the ER and the twDEP methods, respectively, there are two more factors affecting the accuracy of the twDEP method for determining $K_{i4}$ as follows. (i) $K_{i4}$ depends on R^−2^ in the twDEP method, but is independent of R (the particle radius) in the ER method. The size of the test particle should be measured for each individual test in the twDEP measurement for reducing the error. (ii) ER was performed essentially in a local position in the device, and thus the electric field experienced by the particle during the test remains constant in the ER measurement, while the electric field felt by the particle is varying in time as the particle is translating along the electrode array during the twDEP measurement. Although the travelling wave DEP force is essentially constant on a horizontal plane parallel to the electrode array according to the calculation [11], provided it is sufficiently far from the electrode array (about twice the electrode spacing); the time varying electric field effect might induce some errors if the test particle is moving closer to the electrode array, or if there exists inaccuracy in the processes of device fabrication. The discrepancies between ER and twDEP measurements become more obvious when U, or $K_{i}$, approaches zero. It is also noted that the frequency ranges of the ER measurement are much wider than those of the twDEP measurement, $\omega_{4}=10^{4}-4\times 10^{7}$, $10^{3}-4\times 10^{7}$ and $10^{3}-4\times 10^{7}$ for ER measurement, but $\omega_{4}=5\times 10^{5}-4\times 10^{7}$, $6\times 10^{4}-4\times 10^{7}$, and $10^{4}-4\times 10^{7}$ for twDEP measurement when $\sigma_{m}$= 1.2 S/m, 0.1 S/m and 0.01 S/m, respectively, in Figure 7, for example.

Similar results were obtained for lung cancer cells, CL1-0 and CL1-5, as shown in Figure 8, except there exist larger discrepancy at 0.5–0.9 MHz for the lung cancer cells in RPMI solution in Figure 8a,b. This is probably due to the adhesive nature of the cells; CL1-0 and CL1-5 cells adhere to the substrate while the adhesion is very weak for the Colo 205 cell (they slide easily subject to minor fluid oscillation), when they are introduced into the device. Such an adhesive characteristic between cell and substrate is associated with the material property of the cell surface and the substrate, and is not accounted for in Equations (24) and (25). The adhesive effect is pronounced when the cell is at rest, i.e., when U = 0 or $K_{i}$= 0.

In summary, both the ER and twDEP methods can be applied for measuring $K_{i}$, and each of them has its own advantage and disadvantage. The measurement of the twDEP method can be performed much more rapidly than that of the ER method, but the frequency range of measurement is substantially reduced and the accuracy of measurement is lost when $K_{i}$ is near zero using the twDEP method.

The $K_{i}$-spectra for different cells in different medium with different conductivities are re-plotted in Figure 9 for a clear comparison. The results indicate that different cells have different spectra in medium at a given conductivity, and thus the $K_{i}$-spectrum can be served as a physical phenotype of a cell. One can find a certain frequency range that two different cells have different values of $K_{i}$ at a given medium in Figure 9, and thus one can discriminate those two cells based on $K_{i}$, and probably design some apparatus for cell separation and discrimination. The $K_{i}$-spectra or the ER-spectra were always measured in the so-called DEP buffer, with medium conductivity of order 0.01–0.1 S/m, or even lower. The spectra in buffers of physiological strength, as those in Figure 9a, were seldom measured in the literature, but they reflect the cells in their natural environment and could be helpful for understanding certain cell characteristics in their natural state. Hydrolysis did not occur and bubbles were not observed for the present measurements in RPMI solution with medium conductivity at 1.2 S/m. This is due to the low applied ac voltage (2 volts, peak-to-peak) across wide-spacing (100 μm) electrodes at sufficiently high frequency (from 10 kHz–40 MHz). As a comparison, hydrolysis also did not happen in the travelling wave dielectrophoretic pump for human whole blood delivery (with blood conductivity around 0.8 S/m) [41]; the pump was actuated by applying a 5 volts peak-to-peak voltage to the electrode array (with electrode spacing 15 μm) at frequency 1–40 MHz.

## 4. Conclusions

A simple and inexpensive method was proposed, demonstrated and validated for the measurement of the imaginary part of the Clausius-Mossotti factor ($K_{i}$), using electrorotation generated by planar electrodes operated at dual frequency. The method has the advantage that wider frequency range for the $K_{i}$-spectrum and more accurate measurement near $K_{i}$ = 0 can be obtained, in comparison with the method using dual frequency travelling wave dielectrophoresis in the literature. The $K_{i}$-spectra of three human cancer cells were measured, for medium conductivity ranging from 0.01 to 1.2 S/m. Numerical calculations of the dielectrophoretic force and torque in the electrorotation chamber were performed for studying the trapping force and the dielectrophoretic torque exerted on the test particle, which are helpful for designing the device and operation parameters. This study is of academic interest itself, and also finds certain biomedical applications as the $K_{i}$-spectrum can be served as a physical phenotype of a given cell.

## Figures and Tables

**Figure 1 micromachines-11-00329-f001:**
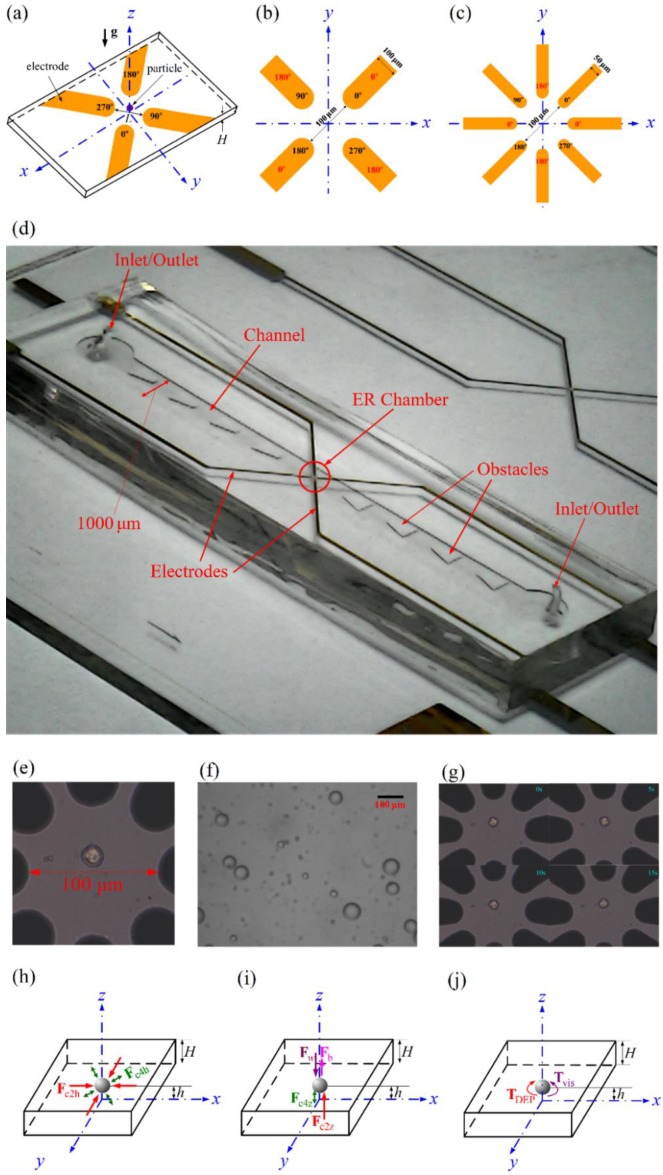
(**a**) Sketch of the electrorotation chamber for measuring $K_{i}$. **g** is the gravity. (**b**) Dual frequency signals were applied to the same electrodes for the four-electrode system. (**c**) Dual frequency signals were applied to different electrodes for the eight-electrode system. (**d**) Device for the four-electrode system. (**e**) A human lung cancer cell, CL1-5, in the test region of the eight-electrode system for experiment. (**f**) Sephadex particles in KCl solution were employed for experimental validation. (**g**) Snapshots from a video for the rotation history of a CL1-5 cell in RPMI solution. (**h**) horizontal force balance of the test particle. (**i**) Vertical force balance of the test particle. (**j**) Torque balance of the test particle for measuring $K_{i}$. The chamber height, *H*, is 80 μm.

**Figure 2 micromachines-11-00329-f002:**
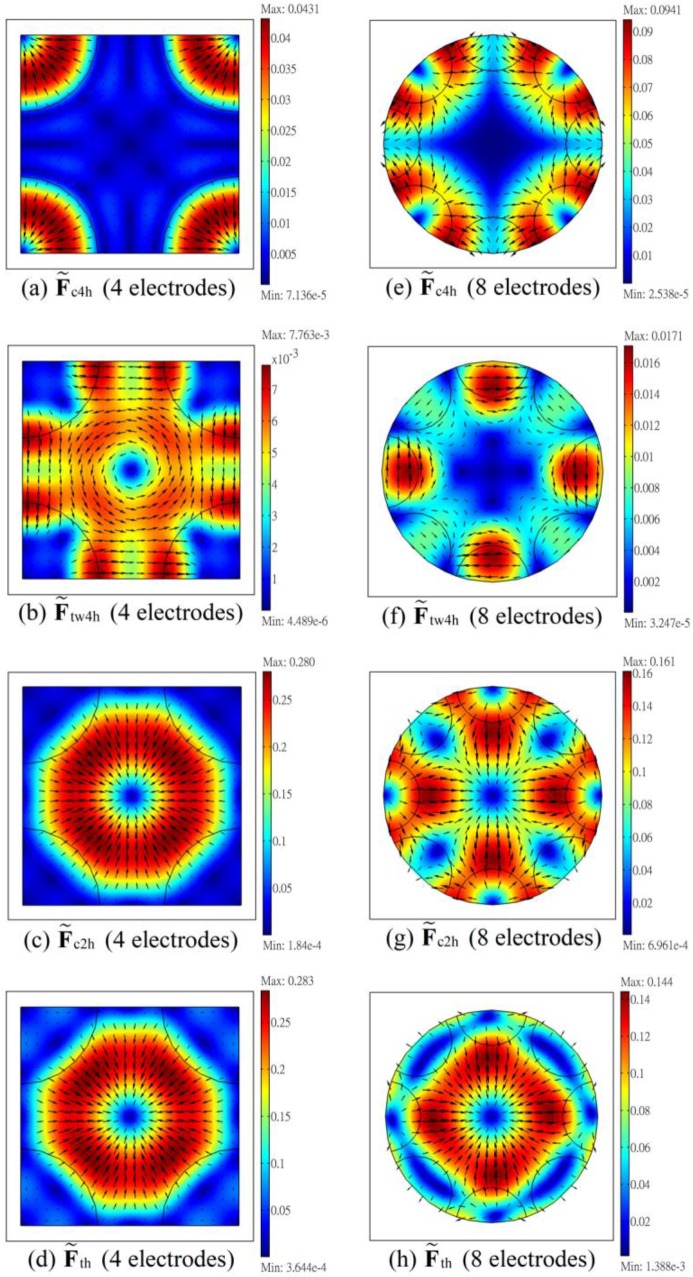
Distributions of dimensionless horizontal forces on the cell on a horizontal plane at height 40 μm in the ER chamber for a typical case under negative dielectrophoresis (DEP). ${\tilde{F}}_{c4h}$, ${\tilde{F}}_{tw4h}$, ${\tilde{F}}_{c2h}$ and ${\tilde{F}}_{th}$ are the dimensionless horizontal conventional DEP force for the four-phase signal, the dimensionless horizontal twDEP force for the four-phase signal, the dimensionless horizontal conventional DEP force for the two-phase signal, and the dimensionless horizontal total DEP force for both signals, respectively. The results are for the case with Colo 205 cell in RPMI solution ($\sigma_{m}$ = 1.2 S/m), calculated using $V_{4}$ = 1 V, $\omega_{4}$ = 20 MHz, $K_{r4}$ = −0.1755 and $K_{i4}$ = 0.0604 for the four-phase signal, and $V_{2}$ = 1 V, $\omega_{2}$ = 0.3 MHz and $K_{r2}$ = −0.256 for the two-phase signal. (**a**–**d**) are the results for the 4-electrode system, and (**e**–**h**) are those for the 8-electrode system. The projections of the electrode edges (part of a circle) on the plane are indicated in the figures, and the tip-to-tip distance of opposite electrodes, s, is 100 μm for all the sub-figures. The forces are normalized using force scale, $2\pi \varepsilon_{m}R^{3}V_{4}^{2}/s^{3}$. The maximum and minimum values are indicated at the upper and lower right corners of each sub-figure.

**Figure 3 micromachines-11-00329-f003:**
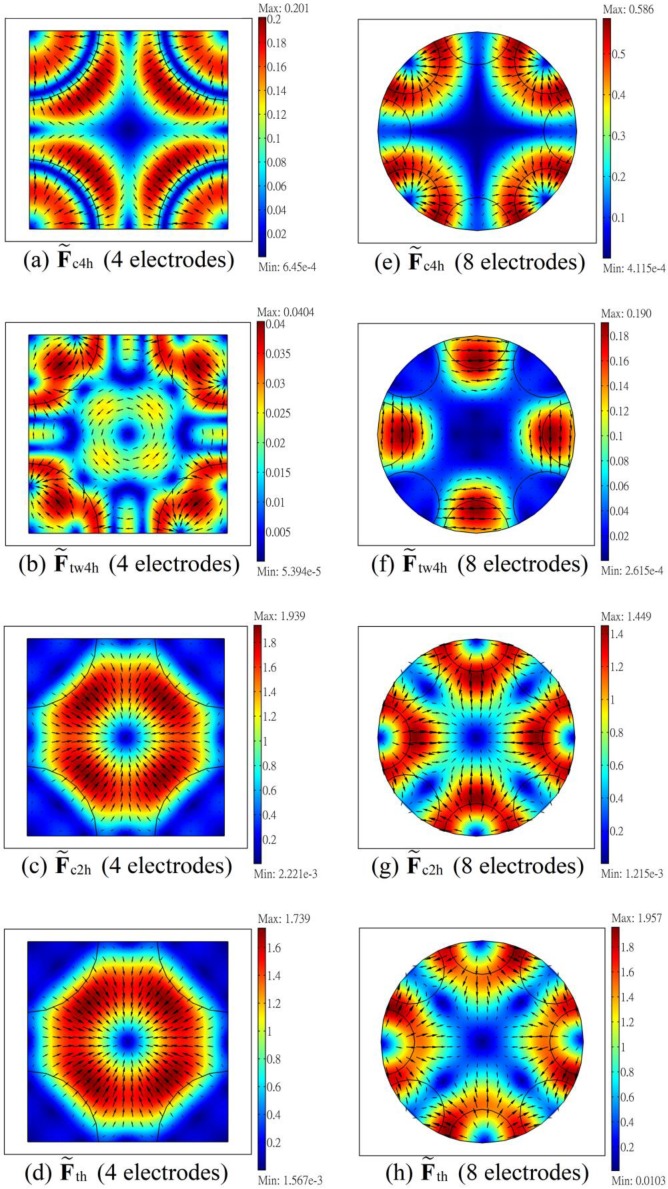
Distributions of dimensionless horizontal forces on the cell on a plane at height 32 μm in the ER chamber for a typical case under positive DEP. The results are for the case with Colo 205 cell in mannitol solution ($\sigma_{m}$= 0.01 S/m), calculated using $V_{4}$ = 1 V, $\omega_{4}$ = 0.1 MHz, $K_{r4}$ = 0.4329 and $K_{i4}$ = 0.254 for the four-phase signal, and $V_{2}$ = 2 V, $\omega_{2}$ = 0.005 MHz and $K_{r2}$ = −0.24 for the two-phase signal. (**a–d**) are the results for the 4-electrode system, and (**e–h**) are those for the 8-electrode system. Graphic notations are the same as those in Figure 2.

**Figure 4 micromachines-11-00329-f004:**
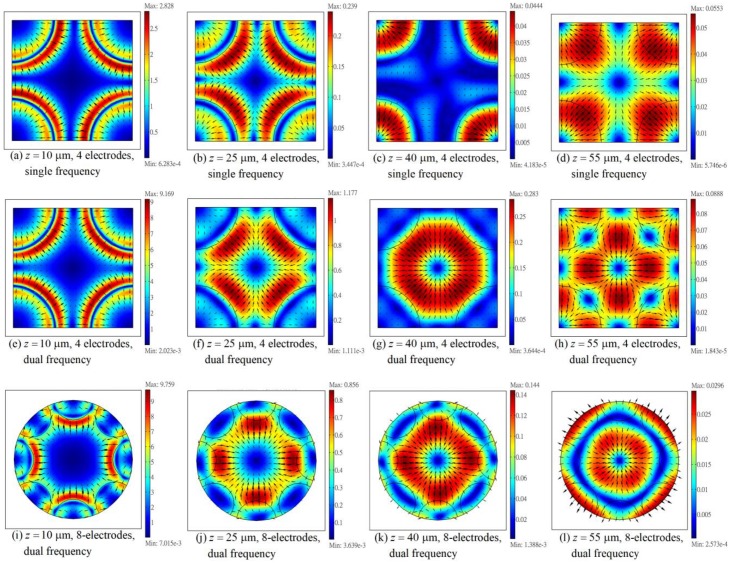
Distributions of the dimensionless horizontal trapping force (${\tilde{F}}_{th}$) on horizontal planes at different heights for the negative DEP case in Figure 2. (**a–d**), (**e–h**) and (**i–l**) are the cases for the four-electrode system with single frequency operation, the four-electrode system with dual frequency operation, and the eight-electrode system with dual frequency operation, respectively. Graphic notations are the same as those in Figure 2.

**Figure 5 micromachines-11-00329-f005:**
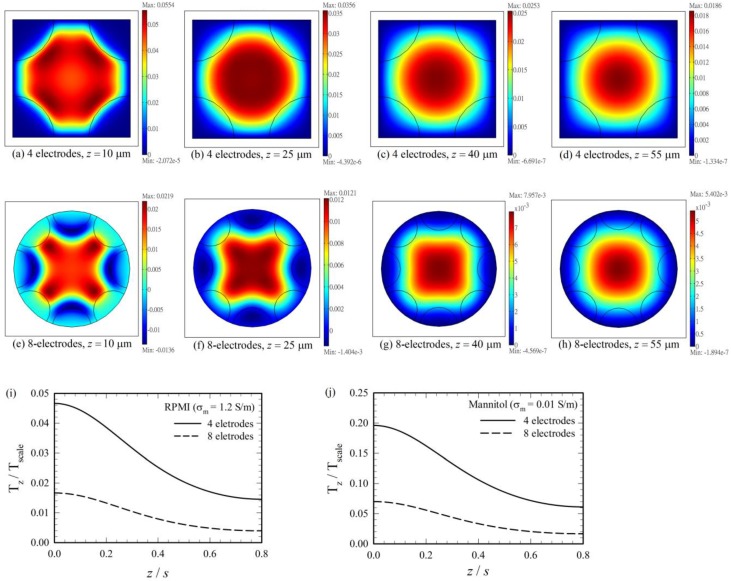
(**a–d**) Distributions of the vertical (z-component) dimensionless DEP torque, ${\tilde{T}}_{z}(=T_{z}/T_{\mathrm{scale}})$, on horizontal planes at different heights of the four-electrode system for the negative DEP case in Figure 2, where $T_{\mathrm{scale}}=4\pi \varepsilon_{m}R^{3}V_{4}^{2}/s^{2}$. (**e–h**) Same as (a–d), but for the eight-electrode system. (**i**) and (**j**) Distributions of $T_{z}/T_{\mathrm{scale}}$ along the vertical axis of the ER chamber for the negative DEP case in Figure 2 and the positive DEP case in Figure 3, respectively. The tip-to-tip electrode spacing *s* = 100 μm.

**Figure 6 micromachines-11-00329-f006:**
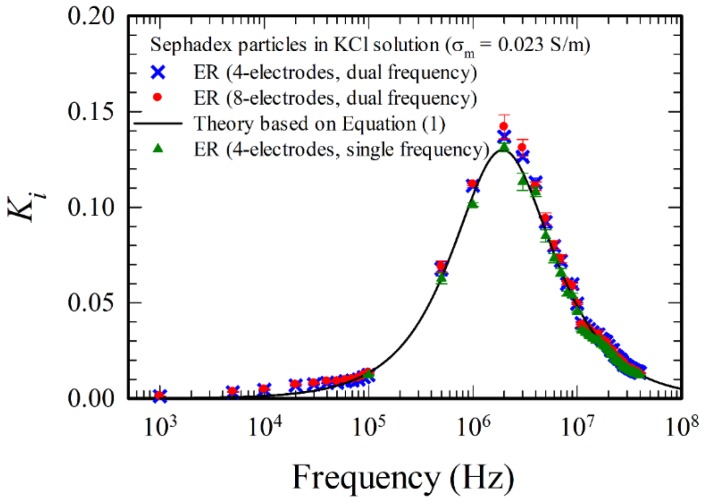
Comparison of the measurements with the theory based on Equation (1), using sephadex particles in KCl solution.

**Figure 7 micromachines-11-00329-f007:**
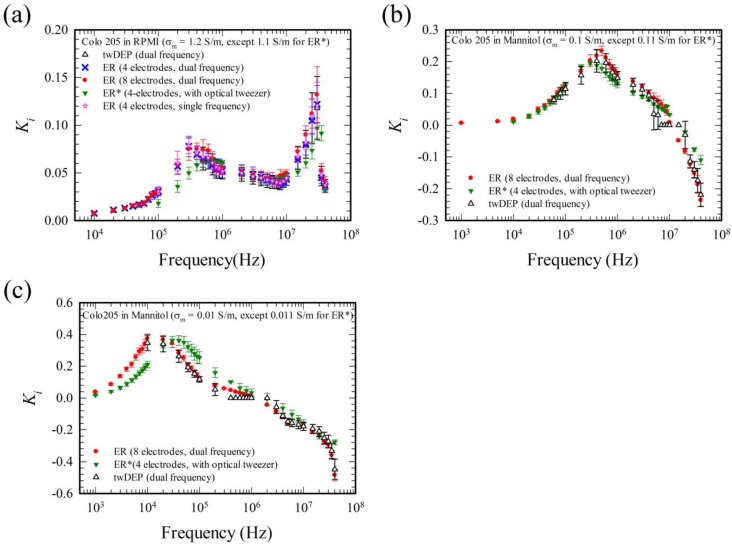
Comparison of the present measurements with experimental results in the literature for Colo205 cells in (**a**) RPMI solution with $\sigma_{m}$ ≈ 1.2 S/m, (**b**) mannitol solution with $\sigma_{m}$ ≈ 0.1 S/m, and (**c**) mannitol solution with $\sigma_{m}$ ≈ 0.01 S/m. ER (4 electrodes, single or dual frequency) and ER (8 electrodes, dual frequency) denote the present experimental results using the four-electrode and the eight-electrode systems, respectively. twDEP (dual frequency) refers to the experimental result in Ref. [11] using twDEP method operated with dual frequency. ER* (4 electrodes, with optical tweezer) refers to the experimental result in Ref. [10] using ER with optical tweezer for particle confinement.

**Figure 8 micromachines-11-00329-f008:**
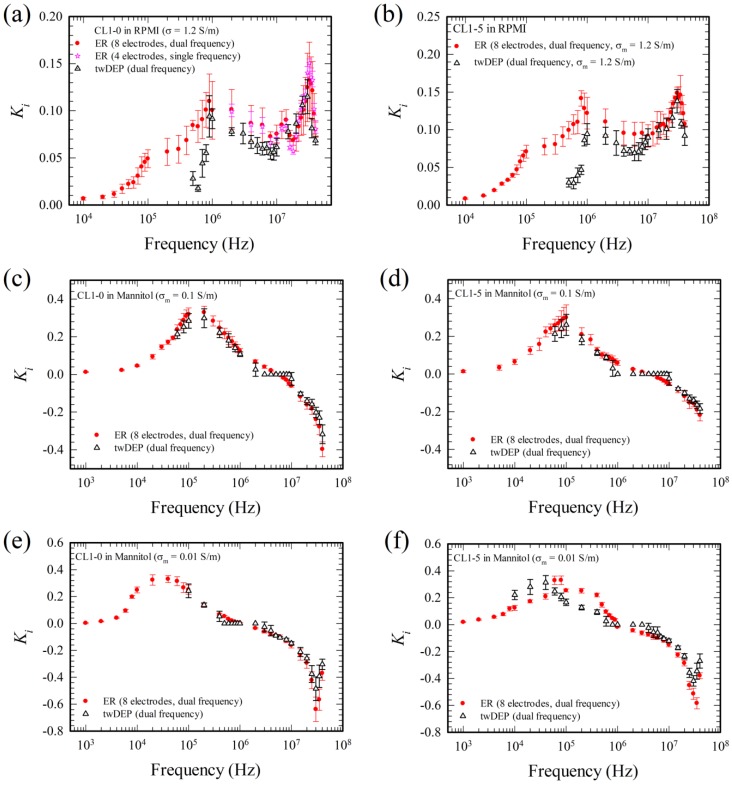
Comparison of the present measurements with experimental results in the literature for CL1-0 and CL1-5 cells, respectively, in (**a**) and (**b**) RPMI solution with $\sigma_{m}$ ≈ 1.2 S/m, (**c**) and (**d**) mannitol solution with $\sigma_{m}$ ≈ 0.1 S/m, and (**e**) and (**f**) mannitol solution with $\sigma_{m}$ ≈ 0.01 S/m. Graphic notations are the same as those in Figure 7.

**Figure 9 micromachines-11-00329-f009:**
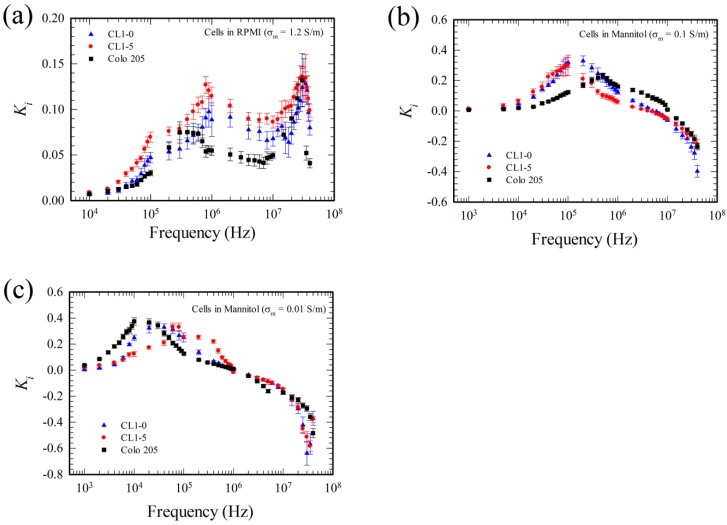
Comparison of $K_{i}$ for various cells. (**a**) Cells in RPMI solution, $\sigma_{m}$ = 1.2 S/m, (**b**) Cells in mannitol solution, $\sigma_{m}$ = 0.1 S/m, and (**c**) Cells in mannitol solution, $\sigma_{m}$ = 0.01 S/m. The results are the present measurements using the eight-electrode system operated at dual frequency.

**Table 1 micromachines-11-00329-t001:** Parameters in the electrorotation (ER) experiment with 8-electrodes under dual frequency operation. The units for $\sigma_{m}$, $V_{4}$ (and $V_{2}$), $\omega_{4}$ (and $\omega_{2}$) and *h* are S/m, volt, MHz and μm, respectively. The values of K_r2_ are adopted from Ref. [5]; those values with * are obtained through extrapolation.

Particles	Medium	$\sigma_{m}$	$V_{4}$	$V_{2}$	$\omega_{4}$	$\omega_{2}$	$K_{r2}$	*h*
Sephadex	KCl	0.023	1	1	0.001–0.1	1	−0.294	32–34
Sephadex	KCl	0.023	1	1	0.1–40	0.05	−0.333	33–39
CL1-0	Mannitol	0.01	1	1	0.001–0.009	0.01	−0.282	36–41
CL1-0	Mannitol	0.01	1	2	0.01–40	0.005	−0.297	31–36
CL1-5	Mannitol	0.01	1	1	0.001–0.009	0.01	−0.179	37–42
CL1-5	Mannitol	0.01	1	2	0.01–40	0.005	−0.168	30–38
Colo205	Mannitol	0.01	1	1	0.001–0.009	0.01	−0.232	35–42
Colo205	Mannitol	0.01	1	2	0.01–40	0.005	−0.240	31–37
CL1-0	Mannitol	0.1	1	1	0.001–0.09	0.1	−0.107	35–40
CL1-0	Mannitol	0.1	1	2	0.1–40	0.05	−0.154	31–36
CL1-5	Mannitol	0.1	1	1	0.001–0.06	0.075	−0.150	36–38
CL1-5	Mannitol	0.1	1	2	0.07–40	0.05	−0.153	30–34
Colo205	Mannitol	0.1	1	1	0.001–0.09	0.1	−0.184	34–39
Colo205	Mannitol	0.1	1	2	0.1–40	0.05	−0.334	31–35
CL1-0	RPMI	1.2	1	1	0.01–0.4	1	−0.156	35–38
CL1-0	RPMI	1.2	1	1	0.5–40	0.3	−0.158 *	33–40
CL1-5	RPMI	1.2	1	1	0.01–0.4	1	−0.164	33–40
CL1-5	RPMI	1.2	1	1	0.5–40	0.3	−0.173 *	34–40
Colo205	RPMI	1.2	1	1	0.01–0.4	1	−0.224	31–36
Colo205	RPMI	1.2	1	1	0.5–40	0.3	−0.256 *	35–41

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
