# Peer review of "Measurement of the Imaginary Part of the Clausius-Mossotti Factor of Particle/Cell via Dual Frequency Electrorotation"

_micromachines, 2020, doi:10.3390/mi11030329_

Round 1

Reviewer 1 Report

The authors have described a clever method of characterizing cells by their electrorotation behavior while being held in negative DEP by a second applied field. I think this is a very well done study and is of great interest to the field.

Minor general grammar editing is needed. For examples line 68 “translation” should be “translational”.

Introduction – needs a little more explanation of what twDEP is and how it differs from normal DEP.

Figure 1 d – Should have a zoomed in microscope image of the actual electrode region in the ER chamber. All the images need scale bars

Line 64 – It is difficult to understand how the conclusion that the ER measurement is better was arrived at. Please clarify the reasoning in the justification at the beginning of the paragraph.

Line 206 – Can you describe the COMSOL model in more detail?

Line 215 – Can you justify why the two phase electric field does not generate any torque on the particle? Is this just for a symmetrical sphere particle or does this apply for asymmetric cells as well? Does the fact that the particle is being held in negative DEP make it harder to rotate than if it were held with laser tweezers? Figure 7 begins to show these differences but a quantitative description of this would be helpful.

Can you add a calculated plot of the torque force generated by the two phase electric field even if it is small? Perhaps a series of graphs like that in Figure 4 can be done with torque between the single and dual frequency situations.

Line 225 – It would be instructive to include image sequences of particles in rotation within the system.

Line 227 – How was the rotation of a sephadex bead measured? Are there distinguishing features on the bead?

Line 253 – The values for the 8 electrode system need to be included as well.

Figure 2 – The vectors arrows are too small to see easily. The image labels are too small to see the subscripts. Describe how these plots were made. If they were generated in COMSOL, a description of the model is needed.

Line 410 – Why was here a 10% difference in medium conductivity between the experiments?

General comments:

Running DEP under high conductance conditions, like the RPMI buffer, often causes electrolysis and bubble formation at the electrodes. It would be good for the authors to include a discussion on this issue. Was it a problem in their setup and if so how they were able to avoid it.

All the error bars in the graphs need to be defined for each figure. The number of samples per data point also needs to be reported for each graph.

Author Response

Please see the attached file entitled "Response to the comments of Reviewer 1".

Reviewer 2 Report

The paper describes a novel dual-frequency electrorotation method to measure the imaginary part of the frequency-dependent Clausius-Mossotti function. A rotating electric field is applied to a quadrupole or octupole electrode configuration in order to measure rotation speed of particles. The authors claim that electrorotation spectra can only be measured when particles experience negative DEP otherwise an additional holding force is required to keep particles in place (as particles that experience positive DEP would simply move out of the four-electrode arrangement towards the edge of one of the electrodes). Thus, they superimpose an additional non-rotating electric field at a frequency that induces negative DEP in the particle and keeps it in place.

While the idea and concept surely merits publication, I have some concerns that should be resolved before it can be accepted:

1. The introduction has essentially no flow and it is very difficult to identify the main issue with conventional electrorotation measurements. As an example, the authors compare the advantages of traveling-wave DEP over ER to measure the imaginary part of the CM function on page 2. But then later, they introduce ER for the first time (L 76: "The so-called electrorotation (ER)...") although they were talking about ER in the previous paragraphs already. I suggest limiting the introduction to ER, only give a very brief comparison to twDEP, and qucikly identify the main problems with conventional electrorotation. Then, introduce the solution which is proposed in this manuscript. Basically, the last paragraph of the intro should outline the contents of the manuscript, best starting with a sentence like "In this manuscript, we propose ..." or something similar. This helps the reader who doesn't want to read the entire manuscript to quickly scan what the study is about. Currently, instead the last paragraph starts with listing studies measuring the real-part of the CM function (which is related to the current study, but has a different context ...).

2. The list of studies about measuring the real part of the CM function misses studies about isomotive dielectrophoresis (for example by Stuart Williams or Tada et al.)

3. Is it actually a problem to keep the cells in the quadrupole electrode arrangement? I have never recorded an ER spectrum in my life (and could not contact associates that have given the short review period). However, skimming through old papers of, for example, Pablo Garcia-Sanchez (on rotation metallic spheres) or Gascoyne (for example the yeast study) I can never find any mention of particles leaving the cage due to pDEP as a problem nor do they state using optical tweezers to affix the cell in the center of the electrode configuration.

Recently Garcia-Sanchez and Ramos published a paper on ER of semiconducting spheres and there is neither a mention of the problem nor of optical tweezers. I urge the authors to compile a more comprehensive list of existing studies using electrorotation and clearly identify at which point it is difficult to keep the particle in place and what causes this; otherwise it is difficult to identify the actually problem statement of the offered solution.

4. I have severe problems figuring out how the superimposed field is supposed to look like. In their study, the authors investigate both fields (the rotating field inducing mainly ER and the non-rotating field that induces mainly DEP) separately and then compare force magnitudes (Fig 2-4). However, how do you even superimpose two signals with two different frequencies on the same set of electrodes? How is this achieved with the voltage source? How does a voltage signal looks like that has two different superimposed amplitudes and frequencies (wouldn't that in effect just lead to some destructive or constructive wave interference or to some odd looking oscillations)? Is it indeed possible to just superimpose two signals and then assume that the ER spectrum is not affected by the presence of the nDEP force? The authors need to prove this more rigorously, either mathematically or experimentally, as currently this is simply assumed to be true.

5. In line with 4: The authors need to present their experimental setup clearer. Please state clearly how the device is fabricated, how the PDMS chamber around the electrodes is fabricated, how the particles are observed, how the rotational speed is measured from the particle, how exactly the voltage is applied (i.e. what device is used), how do you translate the rotation speed to Ki, etc. Currently, the materials section misses the entire setup and protocol.

5.1 What’s the size of the g-25 particles? How is that measured? Are they monodisperse? What is sephadex actually?

5.2 Please describe the COMSOL geometry, model, assumptions, boundary conditions, solutions strategy.

5.3 Give section 2 more structure by introducing sub-sections for setup, materials, simulation, procedure, etc...

6. In line with me having doubts that the ER spectrum is not influenced by the presence of the nDEP force that is applied at a DIFFERENT frequency: The authors should compare their methodology against an established technique. Currently, the authors discuss mostly the comparison against a dual-frequency twDEP device that was described in a different manuscript before. I am aware this might sound unfair, but this study was only cited once and this was a self-citation, consequently, so far I don't want to count this as an established technique (and I dont want to review the dual-frequency twDEP paper to assess the validity of the present manuscript).

6.1 The authors should compare the data of Figure 6, the Sephadex particle, against a conventional ER measurement in a medium of the same conductivity. In addition, on P.7 L228-229, the authors state two different conductivities for the particle (0.007 and 0.023 S/m) and one is according to Ref 28 and the other according to their measurements. Since both values deviate a lot this doesn’t this indicate that their measurement give severely different results? I assume the line in Fig 6 ("Theory based on Eq. (1)") is fitted to the measured spectrum in order to identify the particle's conductivity and is thus not predictive

6.2 There is a significant difference in the maximum of K_i for Colo205 (Fig. 7) between the single frequency and dual frequency measurement. The authors comfortably attribute this to a 10% change in the medium conductivity. However, in panel (a) the conductivity for the single-frequency measurement is lower, but the peak in K_i (and thus the cross-over from nDEP to pDEP) is shifted to right. However, from my understanding, for a two-shell model, at lower conductivity, the first crossover frequency (for the cross-over from nDEP to pDEP) should be shifted to lower frequencies (so in the opposite directions). This would hint that at the SAME conductivity, the discrepancy between both measurements would be even larger. I urge the authors to do the measurement again at the same conductivity and assess the difference between the two measurements at the same conductivity. Same holds for panels (b) and (c) where the conductivity is higher, but the cross-over frequency is shifted to lower values.

6.3 Figure 8, the CL1 cancer lines do not contain comparison against single-frequency ER measurements. This should be included in order to assess the validity of the approach.

6.4 P14 of the manuscript discusses mostly the differences between the dual-frequency ER operation against the dual-frequency twDEP operation. Why is so much weight put on the twDEP operation? This device was not introduced in this manuscript and the large weight put on the twDEP results thus appear odd. As stated again, authors should compare their results against established techniques. Additionally, authors say on line 426 "The agreements are good except the values of Ki when they are near zero". I cannot identify this in Fig. 7 at all. Black and Red/Blue almost always match? Please clarify.

7. The surface plots of the force are difficult to read especially when the arrows are small, bc it is not always clear in which direction the arrows point. Maybe it is better to plot the magnitude of the electric field instead and overlay an arrow plot indicating the DEP force (with force magnitude given by arrow length). Further: P7, L267: "The vectors in each force plot show only the directions (not magnitude) ..." -- I think this is not correct, as the arrows in the plots have different length and I would bluntly assume the length corresponds to the force magnitude.

8. I suggest English proofreading, some sentences are hard to understand or could be improved.

Author Response

Please see the attached file "

Round 2

Reviewer 2 Report

I would like to thank the authors for their thorough revision! A lot of points have been answered. Some issues do remain:

  1. The authors mention that a Ki spectrum and an ER spectrum are not the same. This is true and I am sorry for mixing up the terminology in my first review. Nevertheless, this is just a question of scaling (yes of course the viscosity and the electric field configuration are required to convert an ER spectrum to a Ki spectrum). Since ER spectra are device dependent, comparing those spectra among different studies with each other is almost impossible. I agree. Nevertheless, countless ER studies exist, and while the authors do not explicitly mention the viscosity or show the electric field configuration, they are effectively able to extract the dielectric properties of the target particles from the spectra (i.e., they would be able to plot a Ki spectrum out of their ER data). 
  2. The authors state that not many studies showing Ki spectra exist and this is true but lots of both older and newer studies exist showing ER spectra. I believe some of those should be cited in the manuscript, for example the old works of Pethig or others (interestingly, again, those authors never mention the use of an additional confinement force.. I am not trying to play the devil's advocate here, im rather interested why some authors require those and others don't)

    https://iopscience.iop.org/article/10.1088/0031-9155/37/7/003/pdf

    https://www.sciencedirect.com/science/article/abs/pii/030441659500072J

    https://www.sciencedirect.com/science/article/pii/0005273694901708

    https://ieeexplore.ieee.org/document/1563785

    https://www.sciencedirect.com/science/article/pii/000527369290150K

    https://www.sciencedirect.com/science/article/pii/S0006349591821099

    https://www.sciencedirect.com/science/article/abs/pii/S0925400518315752

    etc.

  3. I am sorry for not finding this earlier, but I only started doing more extensive research after the authors stated that not many studies exist that show Ki spectra from ROT measurements. However, the proposed technique appears not to be novel. The technique of superimposing an nDEP trapping force with a rotating field to measure ROT spectra was (first??) shown by Trainito et al in 2015 in Electrophoresis (10.1002/elps.201400482) to show the influence of electric pulses (electroporation) on the spectra. Then, the technique has also been used to investigate spheroid electroporation (same authors, https://link.springer.com/article/10.1007/s00232-016-9880-7) and by the group of Davalos for the investigation of cancer cells (https://journals.plos.org/plosone/article/file?id=10.1371/journal.pone.0222289&type=printable). Further, there is a paper by the group of Swami that compares sequential nDEP/ROT fields with superimposed nDEP/ROT fields to measure ER spectra while keeping the cells in confinement: Rohani et al, Electrophoresis 2014, 10.1002/elps.201400021. In light of these papers that show the technique already, I believe the authors need to show the novelty and impact of their current study and manuscript, especially in comparison against the existing body of literature. Again, I apologize for not pointing this out in the first revision round!
  4. The paper by Rohani et al. (Electrophoresis 2014) very nicely compares sequentially applied nDEP/ROT applied voltages with superimposed voltages. In their study, they simulate a superposition of the differently applied voltages (which is, as the authors of this study also point out, possible because potentials can be simply superimposed) and look at the direction and magnitude of the resulting field. They find indeed an influence of the nDEP force on the field rotation and find that only in the center of the 4-electrode cage, the field is rotating (which means particle have to be fixed exactly in the center of the 4 electrodes to record adequate spectra). In contrast, when only the rotating field is simulated, the torque experienced by the particle would be very homogeneous in the 4 electrode cage. This is in contrast to the what the authors claim here, which is, that the rotational spectrum is not affected by the superimposed voltage (that achieves the nDEP trapping). Especially, authors derive this mathematically in Eq. (10) and claim that the components 10b and 10c are zero (then, the torque is unaffected as only m4xE4 is left). To me, it is not obvious why 10b and 10c are zero (I did not factor everything out but from a first look I couldn't see it). Can the authors elaborate why they do not find an influence of the linear DEP force on the torque compared to Rohani et al. which do find an influence (and thus recommend sequentially applied nDEP/ROT)?
  5. The net conductivity of polymeric particles in aqueous suspension depends strongly on the particle size. If the Sephadex Particles are polydisperse, then I would expect that their Ki spectrum depends on particle size. In that case, the excellent agreement in Figure 6 is astonishing. Can the authors comment on this?
  6. Apart from the above points: the addition of further results have very much improved the manuscript. I was also not aware that the single frequency measurements for the Colo cells were taken in a different study. Considering this, the agreement is quite good. I never meant to personally or professionally attack the authors when I commented that the twDEP study is not "established" and I am very sorry for expressing myself in the wrong words. I cannot point out any scientific drawbacks of the technique, that was also never my intention. I was simply referring to the fact, that the study is rather new and the technique has not been adopted by the community, thus it is not "yet established" (in the literal sense of the word established).
  7. I know it is cumbersome, but I still recommend a native speaker to proof read the manuscript as the grammar could be improved.

Round 3

Reviewer 2 Report

Thank you for your extensive answers that cleared out my concerns.

I believe the manuscript is now ready for publication.